:ᐤ: PLOS | ONE

# Transboundary movements of foot-and-mouth disease from India to Sri Lanka: A common pattern is shared by serotypes O and C

L. T. Ranaweera[1], W. W. M. U. K. Wijesundara[1], H. S. M. Jayarathne[1], N. J. Knowles[2], J. Wadsworth[2], A. Gray[2], A. M. J. B. Adikari[3], C. K. Weebadde[4], S. D. S. S. Sooriyapathirana[1,5]*

1 Department of Molecular Biology and Biotechnology, Faculty of Science, University of Peradeniya, Peradeniya, Sri Lanka, 2 The Pirbright Institute, Pirbright, Woking, Surrey, United Kingdom, 3 Department of Animal and Food Sciences, Faculty of Agriculture, Rajarata University of Sri Lanka, Puliyankulama, Anuradhapura, Sri Lanka, 4 Department of Plant, Soil and Microbial Sciences, College of Agriculture and Natural Resources, Michigan State University, East Lansing, MI, United States of America, 5 Postgraduate Institute of Science, University of Peradeniya, Peradeniya, Sri Lanka

* sunethuop@gmail.com

**Data Availability Statement:** Relevant data are found within the paper and its Supporting Information files. All FMDV sequence files are

## Abstract

Foot-and-mouth disease (FMD) affects the livestock industry in a transboundary manner. It is essential to understand the FMD phylodynamics to assist in the disease-eradication. FMD critically affects the Sri Lankan cattle industry causing substantial economic losses. Even though many studies have covered the serotyping and genotyping of FMD virus (FMDV) in Sri Lanka, there is a significant knowledge gap exists in understanding the FMDV phylodynamics in the country. In the present study, the VP1 genomic region of FMD viral isolates belonging to serotype C from Sri Lanka and other South Asian countries were sequenced. All the published VPI sequences of serotype C and most of the published VP1 sequences for lineage ME-SA/Ind-2001d of serotype O from Sri Lanka, India, and other South Asian countries were retrieved. The datasets of serotype C and serotype O were separately analyzed using Bayesian, maximum likelihood, and phylogenetic networking methods to infer the transboundary movements and evolutionary aspects of the FMDV incursions in Sri Lanka. A model-based approach was used to detect any possible recombination events of FMDV incursions. Our results revealed that the invasions of the topotype ASIA of serotype C and the lineage ME-SA/Ind-2001d have a similar pattern of transboundary movement and evolution. The haplotype networks and phylogenies developed in the present study confirmed that FMDV incursions in Sri Lanka mainly originate from the Indian subcontinent, remain quiet after migration, and then cause outbreaks in a subsequent year. Since there are no recombination events detected among the different viral strains across serotypes and topotypes, we can assume that the incursions tend to show the independent evolution compared to the ancestral viral populations. Thus, we highlight the need for thorough surveillance of cattle/ruminants and associated product-movement into Sri Lanka from other regions to prevent the transboundary movement of FMDV.

available from the NCBI GenBank Nucleaotide database (https://www.ncbi.nlm.nih.gov) under accession numbers MK390941 to MK390966, KY091301 and KY091302.

**Funding:** NJK, JW and AG received funding from the Department for Environment, Food and Rural Affairs of the UK (Grant No. SE2943), with funding provided from the European Union (via a contract from EuFMD, Rome), and the funding of Biotechnology and Biological Sciences Research Council of the United Kingdom (projects BB/E/I/00007035, BB/E/I/00007036 and BBS/E/I/00007037). SDSSS received funding from National research council, Sri Lanka (NRC grant No: NRC/TO/1410)

**Competing interests:** The authors have declared that no competing interests exist.

# Introduction

Foot-and-mouth disease (FMD) is a highly contagious viral infection that shows significant transboundary movements [1, 2, 3]. Foot-and-mouth disease virus (FMDV) (genus *Aphthovirus*, family *Picornaviridae*) is a small positive-sense RNA virus [4, 5] that affects cloven-hooved ruminants, including cattle [6, 7, 8]. FMDV imposes a massive threat to the livestock industry, causing substantial economic losses [9, 10, 11, 12]. The understanding of the viral phylodynamics is one of the necessities in planning the measures for disease eradication. The knowledge of the demographic histories and tracking the transboundary movements of viral populations using genetic features provide detailed insights into the viral phylodynamics [13].

The VP1 genomic region of the FMDV is one of the common markers used in phylodynamic assessments [5]. The FMDV capsid proteins are encoded by four regions, including VP1 [3, 14, 15]. The protein coded by VP1 is a vital component in host cell attachment and entry [16] and used in characterizing FMDV [17]. Due to the association of type-specific mutations, the VP1 genomic region is commonly used in typing and subtyping of FMDV isolates [18, 19]. The viruses that are undergoing adaptive selection shapes the genetic features enabling the effective host-pathogen interactions [20]. Thus we can use the VP1 genomic region to track the transboundary movements and evolutionary aspects of FMDV.

Many studies suggest that the transboundary FMDV incursions play a dominant role in disease movement [1, 7, 18, 21, 22, 23, 24, 25]. Numerous FMD outbreaks of serotypes O, A and Asia 1 have originated within South Asia [26, 27, 28, 29, 21]. The heterogeneous distribution of FMD within India and the distribution of livestock and their products to neighboring countries lead to transmission of FMDV into the Middle East, South East Asia, and East Asia [30, 31, 32]. FMD has also spread from South Asia to North Africa on several occasions [2]. The characteristic patterns of the epizootic outbreak within a country is visible when introducing FMDV [33, 34, 35, 36]. Commonly, novel incursions often tend to replace old FMDV lineages within a country [37]. In most cases, the spread of FMDV within the country occurs by seasonal outbreaks or severe epidemics in subsequent years after spending quite periods [22]. The lineage expansion events due to recombination or specific mutations can give rise to pandemic situations if the recurrent lineage exists for an extended period [38].

FMDV exists in Sri Lanka since the 19th century [39]. FMD pandemics in Sri Lanka generally appear in every four to six years [40], although the evolutionary dynamics of the patterns of appearance remain ambiguous. To date, only FMDV serotypes O and C have caused outbreaks in Sri Lanka. The FMDV samples from the first outbreak (in 1950) were serotyped and identified as serotype O by a Danish Laboratory. The systemic surveys conducted from 1962 to 1967 and 1977 to 1981 revealed the presence of serotype O in Sri Lanka [41, 39]. Abeyratne et al. (2018) characterized the outbreaks during 1990–2014 [37]. The FMDV infections were reported during 1997–2014, where the numbers were higher in 1997, 1999, 2003, and 2014 [42]. Abeyratne et al. (2018) identified an endemic viral lineage (Srl-97) that caused three outbreak situations in 1997, 1999, and 2000 [38]. In 2014, a massive outbreak was recorded due to the Ind-2001d lineage, sweeping through all the provinces resulting in 58,645 infected cattle and 1,265 deaths [38, 42]. According to the analyses of VP1 and complete genome sequences, the outbreaks of Ind-2001d in Sri Lanka, which occurred in 2013 and 2014, were different from each other. However, FMDV lineages of the 2013 and 2014 outbreaks were closely related to the contemporary Indian FMDV indicating two independent-introductions [43].

FMDV serotype C was first identified in Sri Lanka in 1954 by serotyping of two samples at the Animal Virus Research Institute, Pirbright, England: however, these viral isolates are no longer available. Serotype C was thought to be accidentally introduced from India in 1970, which caused a massive outbreak in Sri Lanka till 1975 [44]. Despite the several attempts on

serotyping and genotyping of the FMDV in Sri Lanka, none of the studies have carried out to understand the transboundary movements and the demographic histories of viral populations considering their genetic features.

Cattle farming is one of the most critical sources of living in the rural areas of Sri Lanka. The development of the cattle industry is a high priority in economic development [45]. FMDV is one of the significant obstacles in developing the cattle industry in Sri Lanka. One of the most effective ways of FMDV eradication in Sri Lanka is the prevention of the transmission of incursions into the country. Moreover, it is essential to understand the FMDV phylodynamics for disease management. Thus the objective of the present study was to find the transboundary movements and evolutionary dynamics of FMDV using genetic data of incursions belonging to O and C, the only serotypes reported in Sri Lanka to date.

## Materials and methods

### Sampling, RT-PCR and sequencing

The VP1 genomic regions of the viral samples collected during the outbreaks caused by FMDV serotype C in Sri Lanka, India, Bangladesh, Nepal, Bhutan, Saudi Arabia, Kuwait and, Tajikistan (former USSR) were sequenced. The vesicular epithelial tissue samples or vesicular fluids were collected from infected cattle. The recommended procedures and the ethical guidelines were followed, as described in Kitching and Donaldson, (1987), for the collection and transportation of the samples [46]. The sampling procedure from the infected cattle was approved by the ethics review committee of the Pirbright Institute under the Animal Scientific Procedures Act (ASPA). The field samples of the present study were submitted to the World Reference Laboratory for FMD (WRLFMD; The Pirbright Institute, UK) during the outbreaks. The total RNA was extracted using the RNeasy kit (Qiagen, Crawley, West Sussex, UK) according to the manufacturer's protocols. The RNA was eluted in nuclease-free water and stored at -20˚C. Soon after RNA extraction, RT-PCR was performed by using the forward primers: *C-1C536*F 5′ TAC AGG GAT GGG TCT GTG TGT ACC 3′ and *C-1C616*F (5′ AAA GAC TTT GAG CTC CGG CTA CC 3′), each with the reverse primer: *EUR-2B52*R 5′ GAC ATG TCC TCC TGC ATC TGG TTG AT 3′ to amplify the VP1 genomic region employing the kits and protocols described in Knowles et al. (2016) [9] for RT-PCR. To confirm the product length of the amplicon, the agarose gel electrophoresis was carried out using the protocols described in Knowles et al. (2016) [9]. The DNA sequencing was carried out using BigDye® Terminator v3.1 Cycle Sequencing Kit (Life Technologies) following the manufactures instructions. Each product was sequenced using the forward primer *C-1C616*F and the reverse primers, *C-1D535*R (5′ ARA GYT CIG CIC GYT TCA T 3′) and *NK72* (5′ GAA GGG CCC AGG GTT GGA CTC 3′). Sequencing was performed using an ABI 3730 DNA analyzer following the instructions provided in Knowles et al. (2016) [9]. The sequences generated during the present study (n = 27) were submitted to GenBank under the accession numbers MK390941 to MK390966, KY091301, and KY091302.

### Phylogenetic analysis

Two separate VP1 sequence datasets were constructed for the serotypes O and C. The dataset for serotype O contained most of the previously published FMD/O/ME-SA/Ind-2001 viral isolates from Sri Lanka and India (S1 Table). For the serotype C dataset, all the sequenced data generated in the present study and all the previously published sequences of FMD/C from India and Sri Lanka (S1 Table) were incorporated. For comparison, the sequence records from all the other topotypes of serotype C were added to make a robust phylogeny. Since the VP1 genomic region is a coding region, the datasets were manually aligned, and the gaps were

added without disrupting the reading frame. The DNA sequence alignment was converted to an amino acid alignment to check for any stop codons in the middle using MEGA 7 [47]. According to the reading frame, the ambiguous ends of the analyzed sequences were trimmed.

The phylogenies for the two datasets were constructed separately following the same procedure. First, coding alignment was used to choose the best nucleotide substitution model that fits the dataset. To achieve a well-fitting model, the dataset was analyzed using multiple methods; namely, Akaike Informative Criteria (AIC) [48], Corrected Akaike Information Criteria (AICc) [49], Bayesian Information Criteria (BIC) [50] and Decision Theory (DT) [51]. The best model was selected using the J model test software [52] in CIPRES Science Gateway [53]. A tree search was implemented in maximum likelihood (ML) criteria in RAxML [54] using the rapid bootstrap algorithm [55] and GTR+GAMMA nucleotide substitution model [56]. The run was iterated for 1000 times and all the bipartitions were used to draw a single topology. Then the bootstrap values were imprinted from bipartition results to the best tree given by RAxML tree search.

Moreover, a Bayesian tree search was carried out in MrBayes [57]. The evolutionary model parameters were incorporated and two hot and cold chains of MCMC were run for 10 million generations. The analysis was set to discard the initial 10% of the trees as the burn-in. From all the trees probed during tree search, the 50% majority-rule consensus tree was taken as the final tree output. To check the robustness of the chain performance and to probe the trees from a stationary distribution, the Effective Sample Size (ESS) was tested using TRACER [58]. All the constructed trees were modified using FigTree [59] for better visualization.

## Construction of haplotype network

The Sri Lankan and Indian FMDV isolates were used to determine the haplotype relationships. The input nexus file was modified to assign the trait labels with respective geographic origins. A Minimum spanning network, Median-joining network [60] and Templeton, Crandall, and Sing (TCS) [61] were constructed separately for both datasets in PopART [62]. The haplotype network with the highest clarity was used to represent the haplotype relationships.

## Detection of possible recombination events

Although a haplotype network can detect particular genetic events, the identification of specific recombination events is difficult. Since the FMDV is an RNA virus, it could undergo recombination to produce novel lineages. Potential recombination events within VP1 were checked using four independent datasets: i) serotype C sequences from India and Sri Lanka (n = 15); ii) serotype C sequences from Sri Lanka and sequences of O/ME-SA/Srl-97 (n = 24); iii) serotype C sequences from Sri Lanka and sequences of O/ME-SA/Ind-2001d from Sri Lanka (n = 22); and iv) O/ME-SA/Ind-2001d sequences from Sri Lanka and India (n = 150) (S1 Table). The Genetic Algorithm for Recombination Detection (GARD) approach [63] was implemented in Datamonkey 2.0 platform [64]. By separately inserting each dataset, the possible recombination events were checked in different serotype/lineage combinations assessed.

## Results

### Phylogenetic analysis

The TrN+I+G model was obtained as the best fit to describe the dataset of FMD/O and TPM1uf+I+G as the best model for FMD/ C dataset (S2 Table). All four criteria used (AIC, BIC, AICc, and DT) yielded the same models separately for two datasets indicating the higher fitness of the models.

The phylogenies created using ML and Bayesian frameworks had almost similar topologies for the two datasets. However, the Bayesian trees had higher resolution than ML trees. The MCMC chains had maximally converged and gave >200 ESS values for all the priors checked. Thus only the Bayesian trees are presented. In the serotype C phylogeny, all the Sri Lankan FMDV isolates that we sequenced and acquired from GenBank fell into a monophyletic group (high bs and pp values) (Fig 1). The south Asian FMDV C strains we sequenced including most of the Indian isolates formed a sister group with the Sri Lankan clade in the Bayesian tree of serotype C. The Bayesian tree constructed for FMDV O/Ind-2001 isolates contained a monophyletic group with Sri Lankan FMDV isolates with higher node support (Fig 2). The GARD analysis did not detect any recombination event within any of the four datasets tested. However, DNA and amino acid sequences analysis detected phylogenetically informative SNPs between Sri Lankan and Indian FMDV.

## Haplotype diversity

The construction of the phylogenetic network provides insights into the relationships between different haplogroups and their movements (i.e., gene flow). Since the virus is more prone to events such as recombination and rapid evolution, it is essential to display the broad picture using the network topologies. The minimum spanning network represents the relationships between Indian and Sri Lankan serotype C viral isolates more clearly than other networks. Due to the availability of a limited number of samples and the fact that FMDV serotype C has not been detected since 2004, the complete haplotypic diversity and the epidemiological patterns in South Asia could not be deduced. However, the gene flow pattern and the direction of the transmission could be recognized via a constructed haplotype network. In the present study, all the published FMD/C sequences in the South Asian region were assessed. For comparison, the FMD/C sequences from the Middle East were used. With these sequences, 38 haplotypes were obtained out of the 40 sequences analyzed. Only shared haplotypes were observed within Sri Lanka and India. We detected 167 parsimony information sites, and the haplotype network had Tajima's D value of 7.759. The D>0 indicates the presence of rare alleles at low frequencies. Thus it is possible to have independent evolutions in Sri Lankan and Indian FMDV populations. We obtained a star topology for the haplotype network separating Sri Lankan and Indian groups with no shared haplotypes. A unique haplotype sampled in India (C/IND/6/71) was the closest nested haplotype with Sri Lankan haplogroup, indicating the possible introduction event from India to Sri Lanka.

The median-joining network with $\varepsilon = 0$ was the best-describing haplotype network with higher clarity for FMD/O/Ind-2001 dataset. With 150 sequences used, we obtained 107 haplotypes, and Sri Lankan and Indian strains did not share common haplotypes. The Sri Lankan haplotypes nested within one haplogroup having a star topology. We observed four more nested haplogroups having the same topology. The Tajima's D value was below 0 (D = -1. 164) for the median-joining network indicating a recent event of population expansion. The distance between haplogroups suggested that the expansion events might be due to recent bottleneck effects or significant changes in the VP1 genomic region.

## Discussion

The VP1 genomic region is one of the widely employed markers in phylogeographic and evolutionary studies of FMDV [5, 19, 65, 6]. Based on VP1 sequences, FMD/C viruses fall into three topotypes; viz. Europe/South America, Asia, and Africa (Fig 1). In the present study, we looked at the evolutionary and phylogeographic perspectives of the VP1 genomic region of FMDV that caused outbreaks in Sri Lanka and India. We checked the transboundary

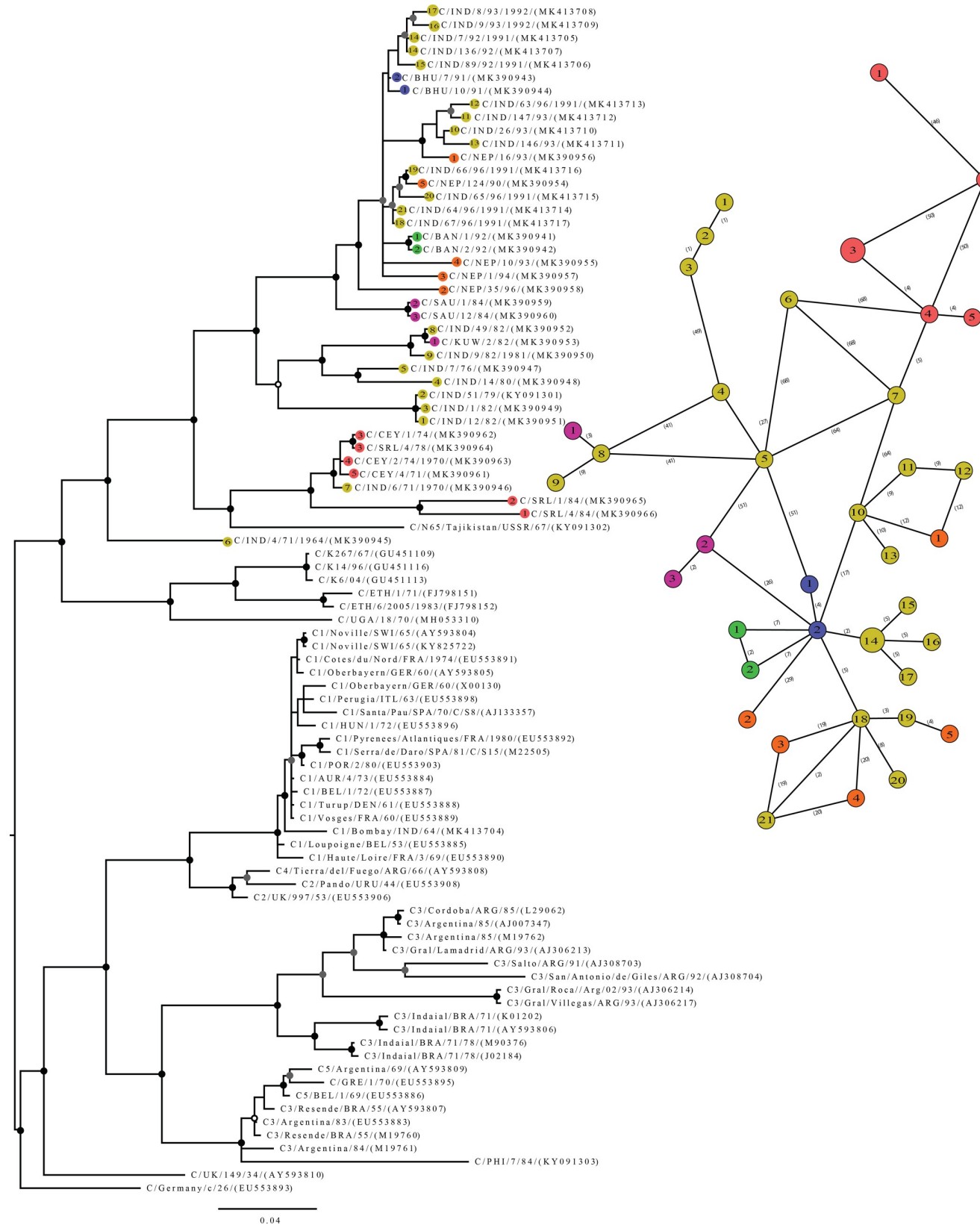

**Fig 1. The Bayesian 50% majority rule consensus tree drawn for the VP1 sequences of the serotype C.** The minimum spanning network drawn for Sri Lankan and Indian isolates are given next to the phylogenetic tree. The tip labels in the tree indicate the haplotypic origin of the sequences (Red: Sri Lanka; Yellow: India; Blue: Bhutan; Orange: Nepal; Green: Bangladesh; Purple: Middle East). Between each haplotype, the distance is given as number of mutations. The scale of the tree is given as substitution per site. In the phylogenetic tree, the node supports are indicated in each node (gray dots: PP>90; white dotes with black outline: bs>70; black dots: PP>90 and bs>70).

movements and underlined evolutionary aspects of the viral outbreaks of Sri Lanka that caused by FMDV serotypes O and C.

In addition to serotype O, which caused several outbreaks in Sri Lanka, three studies reported the existence of the serotype C within the country [8, 40, 44]. For the first time, the present study reveals the dynamics of serotype C virus prevalent in Sri Lanka. All the Sri Lankan viral isolates of FMD/C fell into a single cluster, and they, along with all the South Asian and the Middle East viral strains, fell into the ASIA topotype. However, one of the Indian isolates (C/IND/6/71, collected on 22/09/1970) clustered with the Sri Lankan viruses. According to the nucleotide sequence of the VP1 protein-coding region, there were unique nucleotide substitutions shared by both Sri Lankan isolates and C/IND/6/71. We identified three such nucleotide substitutions at the positions of 255, 408, and 486 (A255G, C408T, T486C) located within the coding region of DE loop, GH loop and β sheet, respectively. Furthermore, deduced amino acid analysis showed the substitution of T151A situated within the GH loop in IND/6/71, which is also shared by Sri Lankan viral isolates (S3 Table). The other Indian viral strains did not share the specific nucleotide substitutions of the Sri Lankan isolates and C/IND/6/71.

The Sri Lankan serotype C sequences were nested into one haplogroup with geographically unique haplotypes. The most closely nested haplotypes for Sri Lankan haplogroup were the Indian haplotypes. The haplotype network had a star topology, and a unique haplotype (C/IND/7/76) was nested as the core haplotype (Fig 1). This pattern reveals the diversification of FMD in South Asia could have started in India. Also, the incursions of FMDV/C in Sri Lanka must have originated from India. The haplotypic relationships indicate that all the FMD incursions in South Asia originated from India and dispersed throughout the region.

In the present study, we did not find any possible recombination of Sri Lankan FMDV. Thus it is possible that the Sri Lankan cluster could have originated from India, where the rapid evolution of Sri Lankan viral isolates formed a genetically distinct form compared to Indian strains through independent evolution. The Sri Lankan viral strains separate into two clades, in which one clade represents the samples collected during 1971–1978, whereas the other clade represents the viral isolates collected during 1984 (Fig 1). These two events could be the lineage expansion events from independently evolving viral strains in Sri Lanka. Our haplotype network also shows that there was no introduction event. Furthermore, the comparison of the nucleotide sequence revealed three nucleotide substitutions at the positions of 33, 39, and 69 (T33C, T39C and A69G) located within the N-terminus. Two nucleotide substitutions at the locations of 409 and 450 (G409A, A450G) located within the coding region of GH loop shared among all Sri Lankan viral isolates found during the 1971–1978 outbreak but not shared with the viral strains collected in 1984 (S3 Table) supporting the fact that the separation could be due to lineage expansion.

Although there have been multiple FMD epidemic situations in Sri Lanka, only a few attempts have been made to characterize the viruses genetically. The sequence data are only available for outbreaks from 1990–2014. Abeyratne et al. (2018) [38] revealed most of the outbreak situations occurred before 2013 due to O/ME-SA/Srl-97 lineage, which is endemic to Sri Lanka. The study showed that Srl-97 endemic lineage is directly monophyletic to "Pak-98" lineage. Therefore, due to lack of data, the linear transboundary movements of the common ancestor of Srl-97 and Pak-98 cannot be deciphered. Moreover, our GARD analysis did not

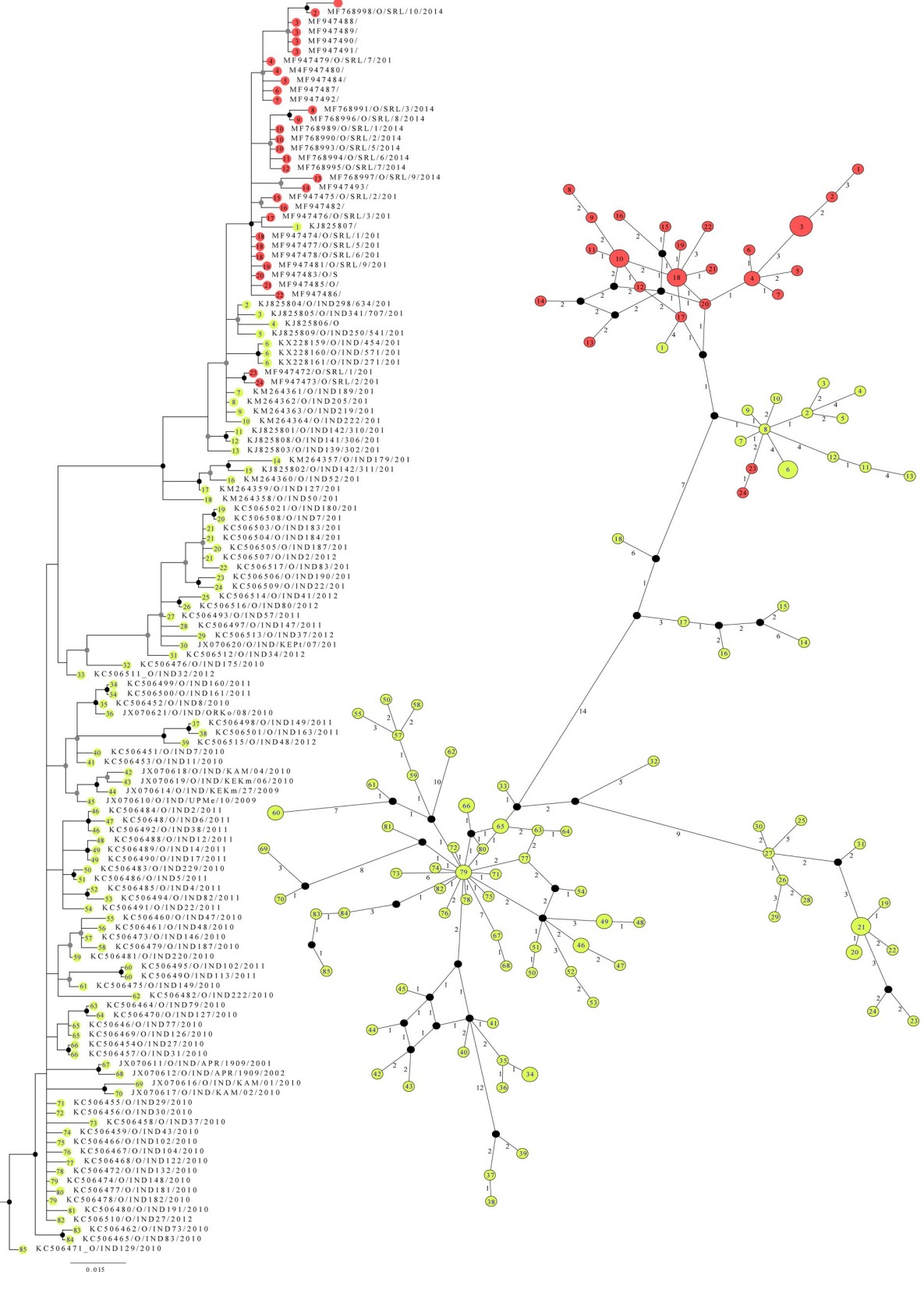

**Fig 2. A clade of the 50% majority rule consensus tree drawn in the Bayesian framework for FMD/O isolates.** We have inserted the median-joining network drawn for Sri Lanka of Indian FMD/O/Ind-2001 strains next to the phylogenetic tree. The tip labels indicate the haplotype origin of the sequences (Red: Sri Lanka; Yellow: India). In the phylogenetic tree, the node supports are indicated in each node (gray dots: PP>90; white dotes with black outline: bs>70; black dots: PP>90 and bs>70). The complete phylogenetic tree is given in S1 Fig. Between each haplotype, the distance is given as number of mutations. The scale of the tree is given as substitution per site.

show any recombination event between FMD/C viral isolates and Srl-97 lineage. Thus it is possible that the Srl-97 lineage underwent independent evolution within Sri Lanka to acquire a unique set of mutations to circulate within the country.

Our analysis of O/ME-SA/Ind-2001 viral isolates of Sri Lanka and India shows a congruent pattern of diversification to serotype C viral isolates. Although the separation in the phylogeny of different groups of strains is low, the median-joining network displayed the grouping of viral strains (Fig 2). The haplotype network shows the diversification patterns of the Ind-2001d sub-lineage. The nested haplogroup of Ind-2001d isolates collected in Sri Lanka was closely related (only four mutations observed) to the haplogroup nested with Ind-2001d isolates collected in India during 2013. Remarkably two isolates collected in Sri Lanka in late 2013 (O/SRL/1/2013 and O/SRL/2/2013) also clustered with this Indian haplogroup. It is likely that O/SRL/1/2013 and O/SRL/2/2013 are closely related to FMD viruses that moved into Sri Lanka from India (Fig 2) [43]. The comparison of nucleotide sequence analysis of the VP1 coding region of viral isolates also supported the proposed hypothesis. We found nucleotide substitution at the position of 78 (G78A) located within the coding region of N-terminus and three nucleotide substitutions at $471^{st}$, $480^{th}$ and $492^{nd}$ positions (G471A, T480C, T492C) within the GH loop shared in Indian viral isolates and O/SRL/1/2013 and O/SRL/2/2013 (S4 Table).

Similarly, O/IND26(54)/2014 formed a unique haplotype that clustered with the Sri Lankan haplogroup suggesting possible limited spread back to India from Sri Lanka. However, it is evident that viruses introduced into Sri Lanka in 2013 and 2014 evolved independently from the ancestral populations (Indian virus) and became enzootic in Sri Lanka. The star topology of the Sri Lankan haplogroup, as well as Indian haplogroup, inferred a rapid expansion of two viral populations. Since the core haplotypes of two disease epidemics are well separated, we can assume that the viruses evolve independently from each other after the incursions occurred.

A phylogeographic study based on the transboundary movement of Malaysia and surrounding countries showed the patterns of FMD movement across the region [22]. The FMDV incursions occurred either annually with quick eradication or introduced virus caused outbreak situation in the subsequent year of the invasion. The latter pattern is much similar to the situation in Sri Lanka and India. From our phylogenetic analysis, it is visible that in serotype C, the disease could have introduced to Sri Lanka in 1971, where the outbreak situation persisted until 1984. During this period, a couple of lineage expansion events were observed, indicating two disease epidemic situations. The lineage expansion could be mainly due to the independently evolving nature of the introduced FMDV in Sri Lanka. Similarly, in serotype O, the virus could have been introduced to Sri Lanka in 2013, although the massive epizootic outbreak started in 2014 and continued throughout the year.

## Conclusion

In this study, a molecular-systematic approach was used to decipher the transboundary movements and the evolutionary aspects of FMDV serotypes O and C into Sri Lanka. It is evident that the FMDV incursions in Sri Lanka were mainly originated from the Indian subcontinent and remained to cause outbreaks in a subsequent year. Moreover, the introduced FMDV

strains tend to show independent evolutions from the ancestral populations, which may complicate the disease eradication. Thus we emphasize the need for policies and surveillance programs to stop the illegal cattle movement into Sri Lanka.

## Supporting information

**S1 Fig. The Bayesian 50% majority rule consensus tree drawn for FMD VP1 sequence of FMDV O isolates from Sri Lanka and India.** The Black dots indicate the nodes with PP>90 and bs>70. The grey dots indicate the node with PP>90. The white dots with black outlines indicate the nodes with bs<70.
(TIF)

**S1 Table. The DNA sequences assessed in the present study.**
(XLSX)

**S2 Table. The nucleotide substitution model parameters for serotype O and C datasets we used in this study.**
(XLSX)

**S3 Table. The nucleotide and amino acid alignments of the serotype C VP1 sequences from Sri Lanka and India.**
(XLSX)

**S4 Table. The nucleotide and amino acid alignments of the FMDV O/ME-SA/Ind-2001d VP1 sequences from Sri Lanka and India.**
(XLSX)

## Author Contributions

**Conceptualization:** L. T. Ranaweera, N. J. Knowles, A. M. J. B. Adikari, C. K. Weebadde, S. D. S. S. Sooriyapathirana.

**Data curation:** L. T. Ranaweera, H. S. M. Jayarathne.

**Formal analysis:** L. T. Ranaweera, W. W. M. U. K. Wijesundara, H. S. M. Jayarathne.

**Funding acquisition:** N. J. Knowles, J. Wadsworth, A. Gray.

**Investigation:** N. J. Knowles.

**Methodology:** L. T. Ranaweera, W. W. M. U. K. Wijesundara, H. S. M. Jayarathne, N. J. Knowles, J. Wadsworth, A. Gray, S. D. S. S. Sooriyapathirana.

**Project administration:** S. D. S. S. Sooriyapathirana.

**Resources:** N. J. Knowles, J. Wadsworth, A. Gray, C. K. Weebadde, S. D. S. S. Sooriyapathirana.

**Software:** L. T. Ranaweera.

**Supervision:** N. J. Knowles, A. M. J. B. Adikari, C. K. Weebadde, S. D. S. S. Sooriyapathirana.

**Validation:** N. J. Knowles, S. D. S. S. Sooriyapathirana.

**Writing – original draft:** L. T. Ranaweera, W. W. M. U. K. Wijesundara, H. S. M. Jayarathne, N. J. Knowles, S. D. S. S. Sooriyapathirana.

**Writing – review & editing:** L. T. Ranaweera, W. W. M. U. K. Wijesundara, H. S. M. Jayarathne, N. J. Knowles, J. Wadsworth, A. Gray, A. M. J. B. Adikari, C. K. Weebadde, S. D. S. S. Sooriyapathirana.

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
