## [Decision Letter · Decision Letter 0]

7 Nov 2019

PONE-D-19-15471

Transboundary movements of foot-and-mouth disease from India to Sri Lanka: a common pattern is shared by serotypes O and C

PLOS ONE

Dear Prof. Sooriyapathirana,

Thank you for submitting your manuscript to PLOS ONE. After careful consideration, we feel that it has merit but does not fully meet PLOS ONE’s publication criteria as it currently stands. Therefore, we invite you to submit a revised version of the manuscript that addresses the points raised during the review process.

The reviewer brings up some important points that need to be addressed, please address the comments and submit a manuscript with track changes on along with a point-by-point response to the reviewer. 

We would appreciate receiving your revised manuscript by November 28, 2019. To enhance the reproducibility of your results, we recommend that if applicable you deposit your laboratory protocols in protocols.io, where a protocol can be assigned its own identifier (DOI) such that it can be cited independently in the future. For instructions see: http://journals.plos.org/plosone/s/submission-guidelines#loc-laboratory-protocols

We look forward to receiving your revised manuscript.

Kind regards,

Douglas Gladue, Ph.D

Academic Editor

PLOS ONE

Journal Requirements:

Reviewers' comments:

Reviewer's Responses to Questions

**Comments to the Author**

1. Is the manuscript technically sound, and do the data support the conclusions?

Reviewer #1: Partly

2. Has the statistical analysis been performed appropriately and rigorously? 

Reviewer #1: Yes

3. Have the authors made all data underlying the findings in their manuscript fully available?

Reviewer #1: Yes

4. Is the manuscript presented in an intelligible fashion and written in standard English?

Reviewer #1: No

5. Review Comments to the Author

Reviewer #1: The paper “Transboundary movements of foot-and Mouth disease from India to Sri Lankan: a common pattern is shared by serotypes O and C” present the sequencing data of foot-and Mouth disease virus VP1 region in order to evaluate the phylodynamics of serotypes O and C.

The data present in this study seems to be interesting in knowing the movement of FMD from the Indian subcontinent to Sri Lanka. Authors extended their study to understanding of genetic variability associated with FMDV VP 1 region to emphasize the effects of this virus on cattle and its impact on associated product movement into Sri Lanka.

During his study authors are mainly focused on retrieving literature data on FMDV and combined several new sequences of serotype C. Majority of those sequences are included Sri Lankan and Indian virus sequences. To infer thee evolutionary and transboundary aspects they used few major software and applications, while the model-based approach is used to detect possible recombination.

However, some issues related to the article should be solved.

Present study authors indicated that they have sequenced viral samples from serotype C. However they haven’t justified why they used only serotype C instead of using both serotypes O and C.

Describing of the sample selection procedure is unclear. Authors are stated that they have used already collected viral samples. But they were not clearly stated how the present study sample is selected. Original sample number and present study cohort selection should be clearly described in the method section. In the present study, authors are newly generated only t 27 sequences of serotype C. Hence clear understand of the sample numbers can justify the present study.

Authors need to expand the description of the structure & phylogenetic significance of VP1 region since that is the major interest of their study.

It will be informative that the authors can include the length of the PCR amplicons and the sequences they used to present study. Shorter the sequence length reduced the precision of the analysis.

It is understandable if the authors may clearly indicate the number of sequences belongs to each dataset that they used to detect possible recombination events (From line #’s 221 to 224)

Some of the references listed in the reference section are not aligned with the journal recommended “Vancouver” reference style

Minor issues

Reference style in some points of the introduction should be revised and corrected

Eg: Line # 106: Reference list should be rearranged

Improving written language style is recommended since some language parts of the article are unclear

6. PLOS authors have the option to publish the peer review history of their article (what does this mean?). If published, this will include your full peer review and any attached files.

Reviewer #1: Yes: Dr Ruwandi Ranasinghe

---

## [Author Response · Author response to Decision Letter 0]

11 Nov 2019

November 11, 2019

Editor,

PLOS ONE.

Dear Editor,

Rebuttal Letter Containing Comments and Answers for the manuscript titled “Transboundary movements of foot-and-mouth disease from India to Sri Lanka: a common pattern is shared by serotypes O and C”

I wish to pay my sincere thanks for reviewing and taking necessary actions to review the earlier submitted manuscript titled “Transboundary movements of foot-and-mouth disease from India to Sri Lanka: a common pattern is shared by serotypes O and C”. I appreciate your valuable effort in this regard to point out areas to improve and strengths in our study which are helpful in improving the quality of the article. This article is of immense importance to the betterment of the Sri Lankan cattle industry. Therefore, with your valuable suggestions, we wish to provide convey our research findings to the FMD research community.

Herewith I am sending the details of actions, answers and comments for each query indicated by the reviewers/editor.

Comments to the Author

1. Is the manuscript technically sound, and do the data support the conclusions?

Reviewer #1: Partly

Answer: The suggested changes were included in revised form of the article. 

2. Has the statistical analysis been performed appropriately and rigorously?

Reviewer #1: Yes

Answer: Thank you for the comment.

3. Have the authors made all data underlying the findings in their manuscript fully available?

medians and variance measures should be available. If there are restrictions on publicly sharing data—e.g. participant privacy or use of data from a third party—those must be specified.

Reviewer #1: Yes

Answer: Thank you for the comment.

4. Is the manuscript presented in an intelligible fashion and written in standard English?

Reviewer #1: No

Answer: The article was edited to correct the language mistakes.

Review Comments to the Author

Reviewer #1

Comment 1: The data present in this study seems to be interesting in knowing the movement of FMD from the Indian subcontinent to Sri Lanka. Authors extended their study to understanding of genetic variability associated with FMDV VP 1 region to emphasize the effects of this virus on cattle and its impact on associated product movement into Sri Lanka. During his study authors are mainly focused on retrieving literature data on FMDV and combined several new sequences of serotype C. Majority of those sequences are included Sri Lankan and Indian virus sequences. To infer thee evolutionary and transboundary aspects they used few major software and applications, while the model-based approach is used to detect possible recombination.

Answer: Thanks for identifying the importance and strengths of this work. 

Comment 2: However, some issues related to the article should be solved. Present study authors indicated that they have sequenced viral samples from serotype C. However, they haven’t justified why they used only serotype C instead of using both serotypes O and C.

Answer: Thank you very much for your comment. I believe a clarification is needed for this question. We wanted to cover FMDV incursions associated with both serotype O and serotype C since those are the only reported serotypes to carry out an FMD outbreak situation in Sri Lanka. (Introduction: Lines 121-122, 149 -150). The sequence data for FMDV Serotype O viral samples collected in Sri Lanka were publicly available and included in the analysis (Materials and methods: lines 180 - 181). Although FMDV epidemic situations are reported for Serotype C, the sequence data were not publicly available. Thus, we collected and sequenced the Serotype C data, in order to collectively analyze the viral phylodynamic for each incursion, as we wished to present a completed and more robust scientific story. 

Comment 3: Describing of the sample selection procedure is unclear. Authors are stated that they have used already collected viral samples. But they were not clearly stated how the present study sample is selected. Original sample number and present study cohort selection should be clearly described in the method section.

Answer: Thanks for identifying this. We have added “We sequenced the viral samples that were collected during the outbreak situations caused by FMDV serotype C in Sri Lanka, India and, a few associated countries”.

Comment 4: In the present study, authors are newly generated only 27 sequences of serotype C. Hence a clear understand of the sample numbers can justify the present study.

Answer: We used sequences of all the relevant FMDV serotype C isolates submitted WRLFMD by the sample collectors. The number of samples for serotype C was turned out be 27. There were no other relevant sequences of serotype C available to have a larger sample size.

Comment 5: Authors need to expand the description of the structure & phylogenetic significance of VP1 region since that is the major interest of their study.

Answer: Thanks for the suggestion. However, I believe the Paragraph 2 of the Introduction justifies the usage of VP1 genomic region and its importance in FMDV phylogenetics. 

Comment 6: It will be informative that the authors can include the length of the PCR amplicons and the sequences they used to present study. Shorter the sequence length reduced the precision of the analysis.

Answer: Thank for pointing this out and making the valuable suggestion. We have remodified the S1 Table including the sequence-lengths.

Comment 7: It is understandable if the authors may clearly indicate the number of sequences belongs to each dataset that they used to detect possible recombination events (From line #’s 221 to 224).

Answer: Thank you for valuable suggestion. The text in the particular section is corrected accordingly as given below.

 “i) serotype C sequences from India and Sri Lanka (n= 15); ii) serotype C sequences from Sri Lanka and sequences of O/ME-SA/Srl-97 (n = 24) ; iii) serotype C sequences from Sri Lanka and sequences of O/ME-SA/Ind-2001d from Sri Lanka (n = 22); and iv) O/ME-SA/Ind-2001d sequences from Sri Lanka and India (n = 150)”.

Comment 8: Some of the references listed in the reference section are not aligned with the journal recommended “Vancouver” reference style.

Answer: Thanks for the comment and the references have been double checked to meet the journal style.

Minor issues

Comment: Reference style in some points of the introduction should be revised and corrected.

Eg: Line # 106: Reference list should be rearranged.

Answer: Thanks, corrected accordingly.

Comment: Improving written language style is recommended since some language parts of the article are unclear.

Answer: Thank you for the suggestion. The relevant changes have been made.

Thank you very much for your consideration and I look forward to hearing back from you with the next step of the publication process.

---

## [Decision Letter · Decision Letter 1]

5 Dec 2019

PONE-D-19-15471R1

Transboundary movements of foot-and-mouth disease from India to Sri Lanka: a common pattern is shared by serotypes O and C

PLOS ONE

Dear Prof. Sooriyapathirana,

Thank you for submitting your manuscript to PLOS ONE. After careful consideration, we feel that it has merit but does not fully meet PLOS ONE’s publication criteria as it currently stands. Therefore, we invite you to submit a revised version of the manuscript that addresses the points raised during the review process.

Please address the few minor comments listed below. 

We would appreciate receiving your revised manuscript by Jan 19 2020 11:59PM. To enhance the reproducibility of your results, we recommend that if applicable you deposit your laboratory protocols in protocols.io, where a protocol can be assigned its own identifier (DOI) such that it can be cited independently in the future. For instructions see: http://journals.plos.org/plosone/s/submission-guidelines#loc-laboratory-protocols

We look forward to receiving your revised manuscript.

Kind regards,

Douglas Gladue, Ph.D

Academic Editor

PLOS ONE

Journal Requirements:

Additional Editor Comments:

Please consider the following suggestions

1) Line # 157: Please indicates what are the "Nearby" Countries

2) Material and Method section: Better rephrase with Passive voice

3)Line # 182: Indicate which type of data set that you used

4)Line # 's 223- 226: Better indicate the individual sequence numbers for each country/Category

5)Line #: Duplication of "Viral"

Reviewers' comments:

Reviewer's Responses to Questions

**Comments to the Author**

1. If the authors have adequately addressed your comments raised in a previous round of review and you feel that this manuscript is now acceptable for publication, you may indicate that here to bypass the “Comments to the Author” section, enter your conflict of interest statement in the “Confidential to Editor” section, and submit your "Accept" recommendation.

Reviewer #1: All comments have been addressed

2. Is the manuscript technically sound, and do the data support the conclusions?

Reviewer #1: Yes

3. Has the statistical analysis been performed appropriately and rigorously? 

Reviewer #1: Yes

4. Have the authors made all data underlying the findings in their manuscript fully available?

Reviewer #1: Yes

5. Is the manuscript presented in an intelligible fashion and written in standard English?

Reviewer #1: No

6. Review Comments to the Author

Reviewer #1: Please consider the following suggestions

1) Line # 157: Please indicates what are the "Nearby" Countries

2) Material and Method section: Better rephrase with Passive voice

3)Line # 182:  Indicate which type of data set that you used

4)Line # 's 223- 226:  Better indicate the individual sequence numbers for each country/Category

5)Line #: Duplication of "Viral"

7. PLOS authors have the option to publish the peer review history of their article (what does this mean?). If published, this will include your full peer review and any attached files.

Reviewer #1: Yes: Dr Ruwandi Ranasinghe

---

## [Author Response · Author response to Decision Letter 1]

6 Dec 2019

Prof. S.D.S.S. Sooriyapathirana

Corresponding Author

December 6, 2019

Editor

PLoS ONE

Dear Editor, 

Submission of the Revised Version of PONE-D-19-15471R1: Transboundary movements of foot-and-mouth disease from India to Sri Lanka: a common pattern is shared by serotypes O and C

We wish to pay our sincere thanks for reviewing the manuscript. Herewith we explain the answers and activities for the queries made by Editor and Reviewer.

Editor’s Comments (EC): 

EC1: Thank you for submitting your manuscript to PLOS ONE. After careful consideration, we feel that it has merit but does not fully meet PLOS ONE’s publication criteria as it currently stands. Therefore, we invite you to submit a revised version of the manuscript that addresses the points raised during the review process.

Answer: Thank you very much for the revisions and herewith we submit the revised version.

EC2: Please address the few minor comments listed below.

Please consider the following suggestions

1) Line # 157: Please indicates what are the "Nearby" Countries

Answer: The ‘Nearby’ countries were included separately in the sentence.

2) Material and Method section: Better rephrase with Passive voice

Answer: The Materials and Methods section was rephrased to passive voice.

3)Line # 182: Indicate which type of data set that you used

Answer: VP1 sequence datasets were used. The sentence was amended to make it clear.

4)Line # 's 223- 226: Better indicate the individual sequence numbers for each country/Category

Answer: We appreciate this comment. The no. of sequences for each country can be easily taken from S1 Table. If we prepare a summary, it would be like the Table given below. Therefore, under the editors permission we wish to keep the current form and guide the readers to S1 Table for sampling details.

Country No. of sequences

Argentina 12

Austria 1

Bangladesh 2

Belgium 3

Bhutan 2

Brazil 6

Denmark 1

Ethiopia 2

France 4

Germany 3

Greece 1

Hungary 1

India 108

Italy 1

Kenya 3

Kuwait 1

Nepal 5

Philipppines 1

Portugal 1

Saudi Arabia 2

Spain 2

Sri Lanka (Ceylon) 28

Switzerland 2

Uganda 1

United Kingdom 2

Uruguay 1

Tajikistan (USSR) 1

5)Line #: Duplication of “Viral”

Answer: We guess the line # has to be 242. The sentence was modified to accommodate the correction.

Reviewers' comments:

1. If the authors have adequately addressed your comments raised in a previous round of review and you feel that this manuscript is now acceptable for publication, you may indicate that here to bypass the “Comments to the Author” section, enter your conflict of interest statement in the “Confidential to Editor” section, and submit your "Accept" recommendation.

Reviewer #1: All comments have been addressed

Answer: Thank you!

2. Is the manuscript technically sound, and do the data support the conclusions?

Reviewer #1: Yes

Answer: Thank you!

3. Has the statistical analysis been performed appropriately and rigorously?

Reviewer #1: Yes

Answer: Thank you!

4. Have the authors made all data underlying the findings in their manuscript fully available?

Reviewer #1: Yes

Answer: Thank you!

5. Is the manuscript presented in an intelligible fashion and written in standard English?

Reviewer #1: No

Answer: The necessary corrections were made, and the Materials and Methods section was changed to passive voice.

6. Review Comments to the Author

Reviewer #1: Please consider the following suggestions

1) Line # 157: Please indicates what are the "Nearby" Countries

2) Material and Method section: Better rephrase with Passive voice

3)Line # 182: Indicate which type of data set that you used

4)Line # 's 223- 226: Better indicate the individual sequence numbers for each country/Category

5)Line #: Duplication of "Viral"

Answer: All these suggestions were addressed before.

7. PLOS authors have the option to publish the peer review history of their article (what does this mean?). If published, this will include your full peer review and any attached files.

Do you want your identity to be public for this peer review? For information about this choice, including consent withdrawal, please see our Privacy Policy.

Reviewer #1: Yes: Dr Ruwandi Ranasinghe

Answer: Thank you! We would also prefer to publish peer review history according to the journal guidelines.

Again we wish to thank the Editor, Reviewer and PLoS ONE staff for your excellent service. I look forward to hearing back from you with the next step of publication.

Thank you!

Sincerely

Prof. S.D.S.S. Sooriyapathirana

Corresponding Author

---

## [Editor Report · Decision Letter 2]

13 Dec 2019

Transboundary movements of foot-and-mouth disease from India to Sri Lanka: a common pattern is shared by serotypes O and C

PONE-D-19-15471R2

Dear Dr. Sooriyapathirana,

We are pleased to inform you that your manuscript has been judged scientifically suitable for publication and will be formally accepted for publication once it complies with all outstanding technical requirements.

With kind regards,

Douglas Gladue, Ph.D

Academic Editor

PLOS ONE
---

## [Editor Report · Acceptance letter]

17 Dec 2019

PONE-D-19-15471R2 

Transboundary movements of foot-and-mouth disease from India to Sri Lanka: a common pattern is shared by serotypes O and C 

Dear Dr. Sooriyapathirana:

I am pleased to inform you that your manuscript has been deemed suitable for publication in PLOS ONE. Congratulations! Your manuscript is now with our production department. 

With kind regards,

on behalf of

Dr. Douglas Gladue 

Academic Editor

PLOS ONE